

# Estimating the uncertainty of sea-ice area and sea-ice extent from satellite retrievals

Andreas Wernecke[1,2], Dirk Notz[1,2], Stefan Kern[2], and Thomas Lavergne[3]

[1]Max Planck Institute for Meteorology, Hamburg, Germany
[2]Center for Earth System Research and Sustainability (CEN), Institute of Oceanography, Universität Hamburg, Hamburg, Germany
[3]Norwegian Meteorological Institute, Oslo, Norway

**Correspondence:** Andreas Wernecke(andreas.wernecke@posteo.net)

**Abstract.**

The net Arctic sea-ice area (SIA) can be estimated from the routine monitoring of sea-ice concentration (SIC) by passive microwave measurements from satellites. To be a truly useful metric, for example, of the sensitivity of the Arctic sea-ice cover to global warming, we need, however, reliable estimates of its uncertainty . We here derive this uncertainty by taking spatial

and temporal error correlations of the underlying local sea ice concentration products into account. Doing so, we find that the observational uncertainties of both sea-ice area and sea-ice extent (SIE) in 2015 are about $300\,000$ km$^2$ for daily and weekly estimates and $160\,000$ km$^2$ for monthly estimates. This is about half of the spread in estimated sea-ice area from different passive microwave SIC products, showing that random SIC errors play at least as much a role in SIA uncertainties as inter-SIC-product biases. We further show that the trend of SIA in September, which is traditionally the month with least Arctic sea

ice is $105\,000$ km$^2$ a$^{-1} \pm 9\,000$ km$^2$ a$^{-1}$ for the period from 2002 to 2017. This is the first estimate of a SIA trend with an explicit representation of temporal error correlations.

## 1 Introduction

In this study we quantify the uncertainty of total Sea Ice Area (SIA) and Sea Ice Extent (SIE) of the northern hemisphere.

We do so by taking into account the spatial and temporal error correlations for propagating uncertainties from the local to the Arctic-wide level for the ESA Sea Ice Climate Change Initiative Sea Ice Concentration Climate Data Record based on the AMSR-E and AMSR-2 instruments at 50km version 2.1 (Lavergne et al., 2019; Pedersen et al., 2017) (hereafter: CCI SIC).

The local area fraction covered by sea ice, called Sea Ice Concentration (SIC), can be inferred at a resolution of a few tens of kilometers from passive microwave radiometers onboard several satellite missions since the 1970s. These SIC estimates do

not depend on daylight, have a small sensitivity to atmospheric conditions and cover most of the polar oceans on a near daily basis. Several passive microwave SIC products exist, and they are valuable tools for many aspects of climate related science,





including operational weather forecasts (e.g. Mironov et al., 2012) and climate monitoring and model assessment (Notz and Marotzke, 2012; Kay et al., 2011; Roach et al., 2020; SIMIP Community, 2020; Stroeve et al., 2007; Ding et al., 2017, 2019).

Based on an analysis of these long-term records, we know that the Arctic sea ice cover is significantly declining for all
seasons (e.g. Stroeve and Notz, 2018). The observed decline of Arctic sea ice has been attributed to a combination of anthroprogenic forcing and internal climate variability, with most studies agreeing that the anthropogenic forcing is the main contributor to the observed loss (e.g. Kay et al., 2011; Notz and Marotzke, 2012; Ding et al., 2017; Fox-Kemper et al., 2021). The vast majority of studies using sea ice as a variable for monitoring and model assessment focus largely or completely on the aggregated measures of SIA and/or SIE (Notz and Marotzke, 2012; Kay et al., 2011; Roach et al., 2020; SIMIP Community,
2020), including the IPCC Assessment Reports (e.g. Fox-Kemper et al., 2021; Gulev et al., 2021, including Cross-Chapter Box 2.2). The need for a robust uncertainty estimate for SIA and SIE products is therefore evident.

The uncertainties in SIA and SIE investigated here stem from uncertainties in the underlying SIC fields. Passive microwave SIC estimates in regions of consolidated ice have typically smaller uncertainties (2% to 8% SIC) than estimates from low to intermediate SIC areas with uncertainties of order of 20% SIC or more (Kern et al., 2019, 2021; Alekseeva et al., 2019; Meier,
35  2005).

Uncertainties in SIC products stem from the interference of atmospheric, ocean and sea ice properties (1), misclassification of surface types (2), the limits in sharpness of the passive microwave measurements (3) and algorithmic uncertainties (4).

(1): The impact of atmospheric interference and roughening of the ocean from wind exposure near the ice edge have been highlithed in Ivanova et al. (2015), and the impact of the surface emissivity variability in general in Andersen et al. (2007).
Tonboe et al. (2021) investigate the sensitivity of several SIC algorithms to geophysical parameters using an emission model. The range of realistic geophysical parameters is based on a multi layer sea ice model forced by re-analysis data. They find that atmospheric variability has generally a small contribution to SIC errors and that, depending on the type of algorithm, either the snow surface density or the snow-ice interface temperature are the largest error sources.

(2): Microwave emissions of wet snow/ice and melt ponds on top of sea ice resemble the emissions of open water more
closely which can lead to misclassifications and hence be a major source of uncertainty for summer melt conditions (Kern et al., 2020; Alekseeva et al., 2019). That being said, the quality of dynamically adapting SIC products, including the CCI SIC, have been shown to be less dependent on the season than others (Kern et al., 2019). Further, thin ice can have passive microwave signature similar to a mixture of thicker ice and open water, adding to the uncertainties in SIC, particularly in summer (Alekseeva et al., 2019).
(3): Smearing effects become important where SIC values vary on scales close to the measurement footprint, for example in the marginal ice zone (Tonboe et al., 2016). SIC algorithms are typically based on several frequency bands with different footprint sizes so that a mismatch occurs in the processing.

(4): The uncertainty in SIE is investigated by Meier and Stewart (2019) by inter-comparison of different SIC algorithms, different processing chains (near-real-time and final product of the NSIDC sea ice index) and by means of sensitivity to
SIC algorithm parameters. They find inter-SIC-product SIE biases to be of order $500\,000$ km$^2$ (in general agreement with the findings of Ivanova et al. (2014) and Kern et al. (2019)), near-real-time versus final sea ice index SIE differences of





order $100\,000$ km$^2$ and the parametric uncertainty of order $50\,000$ km$^2$. However, so far not study exists that has specifically investigated how the local uncertainty in individual grid cell's SIC carries over to the integrated uncertainty of SIA

The CCI SIC product has generally one of the most advanced uncertainty estimates among available products. This uncer-
tainty estimate attempts to represent the four types of sources of uncertainty described above, however, some additional sources of uncertainty cannot be taken into account. Any physical process leading to a bias in the SIC compared to the actual conditions cannot be adequately taken into account by common approaches to construct uncertainty estimates, including the one of the CCI SIC product. Those underrepresented processes include misclassified ice types at the tie points, a possible unaccounted increase in tie point emissions from wet snow (Kern et al., 2020), melt ponds and the influence of weather despite respective
corrections. An additional error source stems from the underlying land mask: if all pixels with some land within them are masked out, the total SIA and SIE will be underestimated (and vice versa).

The knowledge about inter-SIC-product biases in SIA and SIE is crucial, however, it is not suitable as the sole metric to estimate and communicate SIA and SIE uncertainty: For once, the inter-product differences contain constant biases, for example from different land masks, which increase the perceived uncertainty and require in practice a different treatment than
random errors. To focus on inter-product consistency will at the same time not represent uncertainties from sources such as common algorithmic assumptions or errors in the commonly used passive microwave data sets.

To overcome these limitations, we here estimate the uncertainty of a single SIA product based on the unceratinty of the underlying SIC fields. This approach complements the inter-comparison across various products by being based directly on the local SIC uncertainty estimates. Our SIA and SIE uncertainty estimates can accompany the whole product lifespan and can
evolve with changes in product quality and SIC conditions. By representing temporal error correlations we can quantify the reduction in uncertainties from temporal averaging.

## 2 Method

If supplied at all, SIC uncertainties are kept on a grid cell level by the data providers. The analytical propagation of these uncertainties to the aggregated measures SIA and SIE is challenging due to spatial and temporal correlations, computational
constraints (when a full correlation matrix is to be used) and the application of thresholds (for SIE) on SIC fields with dependant uncertainties.

To overcome these issues, instead of an analytical uncertainty propagation we here use a Monte Carlo approach and derive an ensemble of SIC estimates which inhabit correlated error fields. For this approach it is crucial to distinguish between errors and uncertainties: An *error* is the difference between an estimate and the real, typically unknown, value, while the *uncertainty* is
the width of a random variable distribution. In other words, the *uncertainty* is an estimate of the expected absolute amplitude of *errors*. Choosing to represent SIC uncertainties by an ensemble with statistically generated errors allows for easy propagation through even complex calculations: The same calculation (e.g. of the SIE) is performed on each SIC ensemble member creating a frequency distribution for the result. The widths of this frequency distribution can be understood as uncertainty of the result if the following criteria are met: (1) The spread within the ensemble is in agreement with the estimated local uncertainties of the





SIC product, (2) the error correlations of the generated errors are in agreement with estimates of the real SIC error correlations of the product and (3) the ensemble size is sufficiently large.

In the remainder of this chapter we address these criteria, starting in Section 2.1 with an estimate of the error characteristics from CCI SIC data, followed in Section 2.2 by a description of the generation of error fields which are added to the average signal from the CCI SIC data. The last step, described in Section 2.3, is to test the generated samples to be a good representation
of the SIC uncertainty and hence to fulfill the first two criteria above.

## 2.1 CCI SIC error characteristics

The error correlations are assumed to be stationary in space and time with one characteristic length scale in space and one in time. Correlations are therefore assumed to solely dependent on the (space-time) distance between two locations.

### 2.1.1 Spatial correlation

The estimate of the spatial correlation length scale used here is based exclusively on Kern (2022), described in Kern (2021). In Kern (2021), the CCI SIC is used and locations of high concentration pack ice (SIC>90%) are selected. At these locations circular discs are used to calculate, within a 31 day temporal window, the correlation between the SIC or CCI SIC uncertainty estimate between each disk, with increasing radius, and the center point. Exponentially decaying functions are fitted to the correlation as a function of distance to the center. The e-folding distance, i.e. the distance at which the fit reaches 1/e, is saved
restricted to the range of [20 km,1000 km], in steps of 5 km. There are two types of correlation length scales discussed in Kern (2021), depending on which variable is used in the method outlined above, namely those based on the total_standard_error variable and those based on the sea-ice concentration variable, which will be introduced in the following.

The total error correlation length results from the described processing when using the total_standard_error variable (renamed to total_standard_uncertainties in newer versions) of the CCI SIC product. The total error correlation length therefore
describes whether the amplitude of uncertainties is correlated but not whether the errors that these uncertainties describe are correlated. An example for this would be a process which creates independent noise on a persistent spatial scale. In this case the amplitude of the noise would have a typical length scale but the errors would nevertheless be uncorrelated. The total_standard_error variable is largely based on the maximum SIC minus minimum SIC of a moving 3×3 grid-cell box (corresponding to 150 km × 150 km for the 50 km resolution product) which is used to include the dependency of the smearing
effect on local SIC variability (Tonboe et al., 2016; Lavergne et al., 2019).

The sea ice concentration error correlation length (hereafter: SIC error length), in contrast, results from the described processing when using the raw SIC values themselves, including values outside the [0%, 100%] range. Analysing these untruncated SIC values shows that they regularly reach up to 110% SIC, indicating that product SIC errors, even for pack ice regions, are of order 10% SIC, since SICs above 100% are physically impossible. By assuming a symmetric error distribution it follows that
all SIC values above 90% can originate from fully ice covered regions, which informed the >90% criterion of Kern (2021). The analysed correlations are in fact caused by a combination of situations where the actual SIC is slightly below 100%, leading to a reduction in observed SICs, and errors in the observations which are not reflecting the real world situation. Here we assume



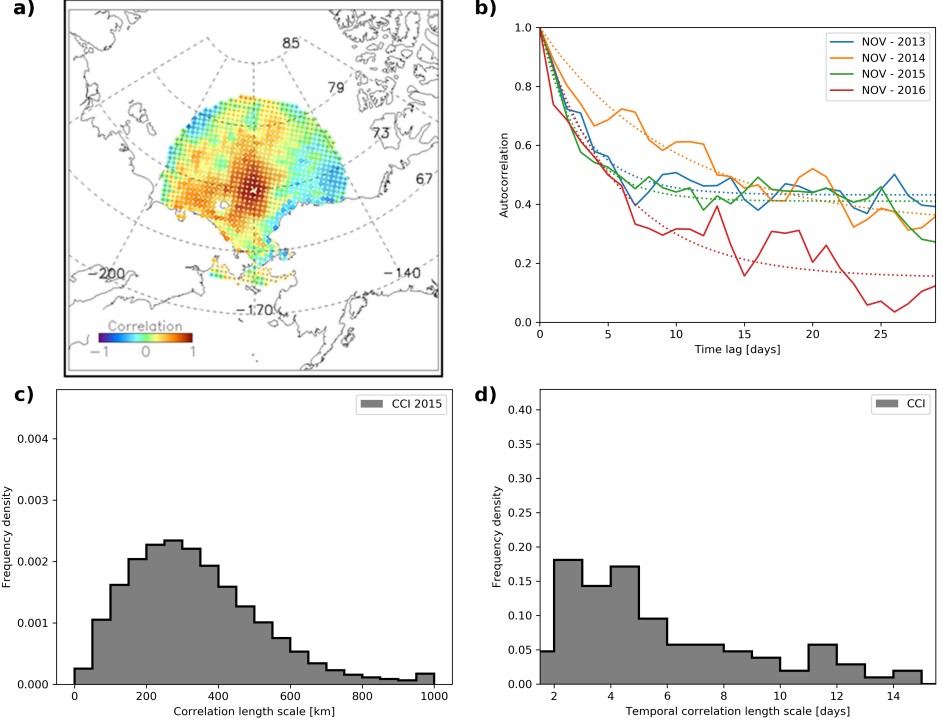

**Figure 1.** Spatial (left) and temporal (right) correlation characteristics. Example of the correlation with a selected location (center) near Wrangel Island on January 26 2010 (a), taken from Kern (2021), and frequency distribution of resulting spatial sea-ice concentration error correlation length scale for the northern hemisphere and the whole year 2015 (c). Examples of the auto-correlation values (solid lines) for November months from 2013 to 2016 with minimal RMSD fit (dotted lines) (b) and resulting frequency distribution of the temporal correlation length scale for the whole years from 2013 to 2016, inclusive (d).

that locations with real SICs very close to 100% dominate in the analysis for the SIC error length so that the SIC error length is a good measure of the product's error correlation (see Discussion for more information on this assumption). In other words,

for a real SIC of 100%, the correlations in the derived SIC product originate solely from the retrieval errors. It is this error correlation which is required for the statistical error generation proposed here which is why we will focus on the SIC error length in the following.

Figure 1c shows that, for 2015, the spatial error correlation length, as provided by Kern (2021), peaks at around 300 km. The generated samples will be designed to echo this distribution. For a lack of other information, we assume that this error

correlation length can be applied to all locations, independent of the local SIC value.

### 2.1.2   Temporal correlation

We derive the temporal error correlation length (or: duration) in a way largely consistent to the processing of the spatial SIC error length in Kern (2021). Using the untruncated SIC fields of the CCI CDR we check each month for locations at which the





daily SIC is not falling below 90% SIC. Based on all these locations we derive a monthly autocorrelation time-series and find

the minimum RMSD fit of an exponentially decaying function, $c_t(\Delta t)$, to it (Equation 1, Figure 1b). The e-folding value, $\ell_t$, of this fit is used as a measure for the temporal error correlation length. The seasonal cycle and potential trends are not removed from the SIC data set for this processing because they are expected to have negligible impact on timescales of several days to weeks. However, we do allow $c_t$ to converge to a floor level $c_f$ different from zero:

$$c_t(\Delta t) = (1 - c_f) \cdot \exp\left(\frac{\Delta t}{\ell_t}\right) + c_f \tag{1}$$

where $\Delta t$ is the time-lag and $\ell_t$ is the temporal error correlation length. The addition of a floor level correlation results in much better fits of $c_t$ to the autocorrelation data (Figure 1b) which improves the representation of the initial drop in autocorrelation of interest here.

## 2.2 Monte Carlo modelling

We create a Monte Carlo ensemble in order to propagate the CCI SIC uncertainty estimates with the previously found spatial

and temporal error correlations to the SIA and SIE estimates. The spread within the SIC ensemble represents its correlated uncertainties and therefore the ensemble-spread in the resulting SIA or SIE estimates provides an estimate of the propagated uncertainty.

The generation of ensemble members with correlated random errors consists of the following four steps: (1) Independent white noise is generated in the whole domain by a numerical random generator. The noise is generated for the whole time

period and hemisphere at once to avoid discontinuities in the final error fields. (2) A three-dimensional Gaussian low-pass filter with sigma values of 5 days in the time dimension and 288 km in the two space dimensions (compare Figure 1c and d) is applied to the independent noise to remove higher frequency components. Note that these values are informed by, but not equivalent to the analysed error correlations shown in Figure 1. The quality of the generated noise will be evaluated in the following section. Two alternative types of filters have been applied for comparison which have limited impact on the results

(see Appendix). (3) The amplitude of the filtered noise field is normalized to have a standard deviation of one. (4) the noise field is multiplied with the total_standard_error variable of the CCI SIC product. All noise realizations are added individually to daily fields of the SIC product from which the high-frequency variations have been filtered. The CCI SIC product contains errors itself; the objective here is to replace these errors by statistically generated ones. Therefore we remove high frequency SIC variations by using the same Gaussian filter on the SIC data, as is used to create errors fields. Without this step we would

add generated error fields to a SIC field which already contains the same type of error, so that the resulting fields would by default show stronger spatio-temporal variability.

## 2.3 Quality of generated noise

The generated SIC fields should fulfill the two basic quality measures mentioned before: How well is the local spread within the ensemble reflecting the uncertainty estimates of the CCI SIC product? And how well does the generated ensemble reproduce





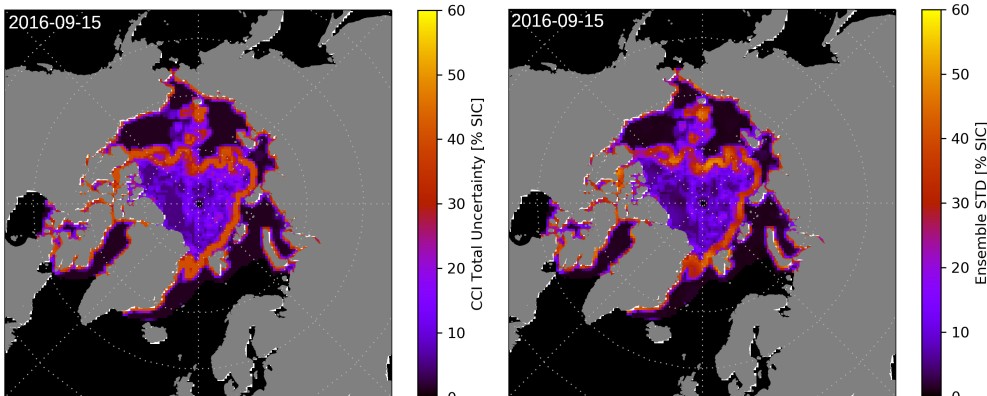

**Figure 2.** Example of uncertainty estimates as provided by the CCI SIC product variable *total_standard_error* (left) and the standard deviation of 100 members of the statistically generated ensemble using the Gauss filtering approach.

the spatial and temporal error correlation characteristics of the original product? If both criteria are met, we have shown that our synthetic errors are a good approximation for the inherent product errors with statistically generated errors.

### 2.3.1 Local uncertainties

To examine that the first quality measure is met, we compare the spread in the generated ensemble with the uncertainty estimate by the data providers (Figure 2). We find that indeed, the ensemble spread is very similar to the *total_standard_error* variable

which is the combination of the algorithmic uncertainty and the smearing uncertainty, representing one STD in percentage points of SIC. It is outside the scope of this work to derive individual error characteristics for those contributing uncertainties. To be clear, we do not asses the quality of the local CCI SIC uncertainty estimate here but focus on creating a statistical representation of this product.

### 2.3.2 Error characteristics

The spatial and temporal error characteristics of statistically generated ensemble members is similar to the original CCI SIC product as well (Figure 3). For Figure 3 we use the same approach to derive spatial and temporal error correlation length as described above on one noise realization to be compared with the CCI SIC characteristics. It can be seen that not only the average correlation length scales agree between the CCI SIC and statistically generated ensemble members, but that furthermore the width of the distributions are consistent.

In summary, we have generated a statistical ensemble of SIC time-space fields which are centered around the CCI SIC product while the added noise is in excellent agreement with the local CCI SIC uncertainty estimates and the estimated temporal and spatial error correlations. We can therefore in the following use the generated ensemble to quantify uncertainties in the SIA and SIE.





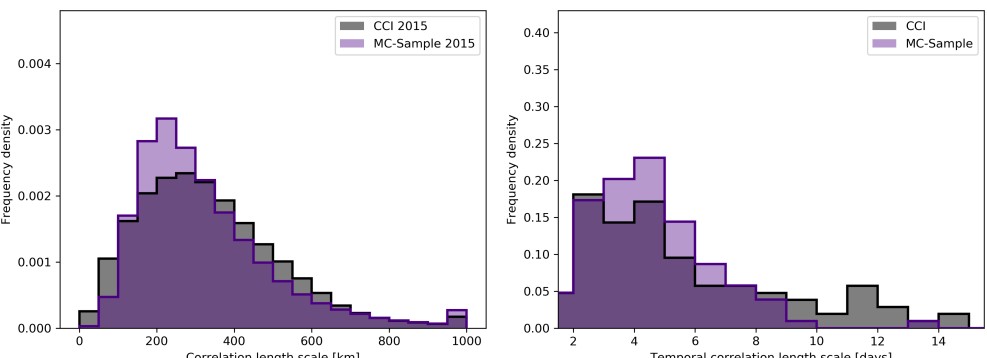

**Figure 3.** Comparison of the CCI SIC error characteristics, as shown before, with the characteristics of one statistically generated ensemble member. The shown spatial distributions for 2015 have a mean of 333 km (CCI) and 322 km (sample) and a median of 305 km (CCI) and 280 km (sample). The shown temporal error distributions have a mean of 5.8 days (CCI) and 4.5 days (sample) as well as a median of 4.7 days (CCI) and 4.2 days (sample).

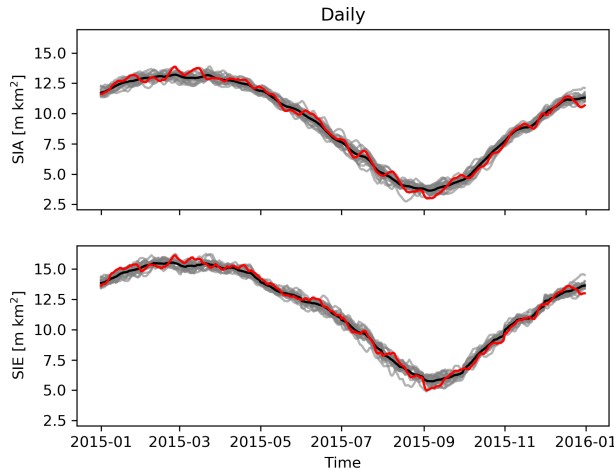

**Figure 4.** Daily Arctic SIA and SIE ensemble of 20 SIC ensemble members for the year 2015. One member is highlighted (red) to illustrate the temporal correlation of the time series. The ensemble mean is shown in black

## 3 Results

To derive SIA and SIE uncertainties we repeat the calculation of daily SIA and SIE values for each ensemble member individually (see Figure 4 for an example). Note that the temporal correlation in the SIC errors results in increased smoothness over time in the SIA and SIE variability of ensemble members compared to temporally independent noise (see Figure 4). The errors in SIA and SIE can neither be well represented by a constant bias nor by temporally independent noise, which highlights the value of our approach to statistically model the underlying SIC error.




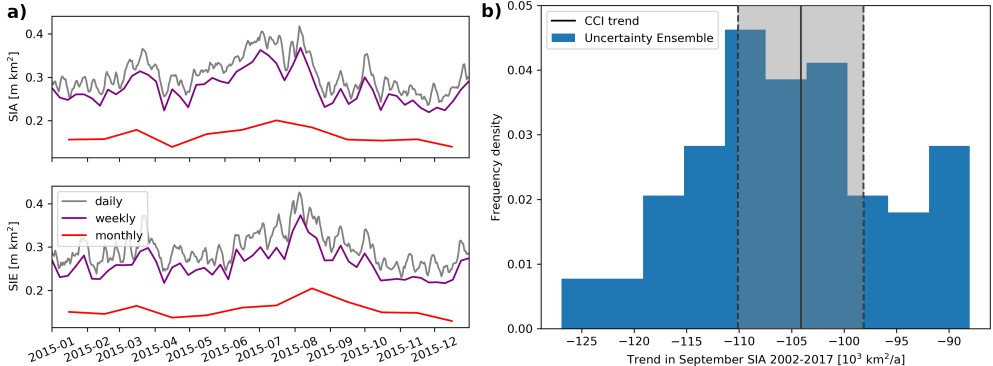

**Figure 5.** a: Standard deviations of SIA and SIE derived from an ensemble of 100 SIC ensemble members generated with a Gauss filter. Shown are daily (black) values as well as weekly (purple) and monthly (red) averages. b: Frequency distribution of 2002 to 2017 September SIA trend from the same type of ensemble (blue) with the trend from linear regression based directly on the CCI product for comparison (black line) and one standard error of the trend (gray shade).

In a next step we derive the weekly and monthly SIA and SIE estimates from the daily time-series and calculate the ensemble STD (Figure 5a). The SIA uncertainty in 2015 is 306 000 km² for daily, 275 000 km² weekly, and 164 000 km² for monthly estimates. The SIE uncertainty in 2015 is slightly smaller with 296 000 km² for daily, 261 000 km² weekly, and 156 000 km² for monthly estimates.

This shows that weekly averages have nearly the same uncertainty as daily estimates which is due to the typical temporal
error correlation of 5 days. In general, the uncertainty of daily SIA and SIE values is highly dependent on the spatial error correlation and the level of reduction by temporal averages is determined by the temporal correlation (see Appendix). The reduction in errors by weekly averages is small because errors do not cancel efficiently because of their correlation on a similar temporal scale. For monthly estimates, in contrast, the uncertainty is reduced by a factor of about two. Figure 5a, indicates a small increase in SIA and SIE uncertainties in summer (JJA) compared to the rest of the year.

We address the sensitivity of our uncertainty estimates on the SIC error correlation length scales by repeating the ensemble generation for realistic lower-end and upper-end correlation length scales. The difference between those extremes in SIA and SIE uncertainties is about 80 000 km² for both quantities (see also Table A1).

A quantity of large interest is the trend is September SIA and SIE because September is the month which typically contains the yearly sea ice minimum and is one of the months with the fastest observed decline in sea ice (Stroeve and Notz, 2018).
We derive the linear trends of the ensemble members by a minimum RMSD fit to the September daily SIA estimates from 2002 to 2017 and analyse their statistical distribution (Figure 5b). The ensemble of SIA trends for this period has a mean of 105 000 km² a$^{-1}$ with one standard deviation of about 9 000 km² a$^{-1}$. The standard error of the trend, estimated directly from the CCI SIC data, is of similar size. Considering the different nature of these uncertainty estimates with comparable values supports our confidence in the ensemble error representation and the provided local SIC uncertainty estimates of the CCI SIC
product.



## 4   Discussion

The uncertainties represented here can be understood as an improved representation of the total_standard_error variable pro-
vided with the ESA CCI sea-ice concentration product. We fully rely on the experience and extensive validation efforts of the
data providers to quantify the local uncertainties (Kern and Timms, 2018). As mentioned before, these uncertainty estimates
summarize the impact of several sources of uncertainty (see introduction). However, biases in particular are not represented
here, for example those from the applied land mask, which would further require a separate statistical treatment.

The assumption of homogeneous and purely radial error correlations is a strong simplification. This is because some sources
of errors are expected to play a stronger role for specific conditions. This includes the land spill-over, which originates from
a strong contrast between microwave signatures from land and the ocean while the signatures of land and sea-ice are very
similar. The passive microwave sensors permit a blurring of this sharp contrast, leading to a contamination of measurements
near the coast with land signatures (Parkinson, 1987; Cavalieri et al., 1999) that can confuse the SIC retievals. This is leading
to a potential SIC overestimation, in particular for low ice conditions near the coast where the contrast between ocean and land
emissivities is largest. Filters to reduce the land spill-over effect exist and are used also in the CCI SIC product (Lavergne et al.,
2019). Nevertheless, this effect is creating increased uncertainty in some cases and is likely to create correlated errors along
the coast which lose correlation much quicker in offshore direction.

Another error source with likely non-circular error correlation are tie-points. Tie-points act as reference values for regions
of consolidated ice, sometimes split into consolidated first-year ice and consolidated multi-year ice, and open ocean. Errors
in tie-points are expected to create errors at all locations with conditions similar to the tie-point conditions. Therefore error
correlations are expected to be higher within groups of locations with high fractions of one class. In other words, since the
ocean and sea-ice tie-points are defined independently from each other, one would expect that the errors at an open water
location show only limited correlation with errors at a nearby sea-ice location despite the distance being within the correlation
length scales.

Despite these mechanisms we use strictly circular correlation pattern in this study, based purely on the distance between
two locations. The existence of non-circular correlation pattern is further supported for example by Figure 5 in Kern et al.
(2021). However, taking these into account requires additional research to quantify the cause, abundance and impact of those
non-circular components . In general an increase in error correlations at locations which high uncertainties, such as the coast
and marginal ice zone, would correspond to larger uncertainties in the SIA and SIE.

Another assumption we rely on is that the error characteristics derived from nearly 100% SIC are applicable for all ice
conditions. A similar analysis for intermediate SIC is not possible because variations in real SIC and SIC errors cannot be
distinguished. For conditions close to 0% SIC at locations close to the ice edge, the approach of Kern (2021) could be applied
in principle but there is a larger chance of ice floes passing through the area, again making the distinction between errors and
real SIC variations difficult. For high SIC areas, leads or other openings in the ice can have the same effect but typically close or
freeze over within days. Leads covered with thin ice can cause passive microwave products to show reduced SIC values, which
we consider an error in the SIC estimate. Therefore we want over-frozen leads to be represented by the error ensemble. For a





better understanding of error correlations a large set of high quality reference data would need to be assembled and analysed for passive microwave SIC error characteristics, which currently does not exist.

Comiso et al. (2017) compare four different SIC products and find the decline in annual minimum SIA to be $79\,300\,\mathrm{km^2 a^{-1}}$ on average for the period from 1979 to 2015. This is smaller than the decline of about $104\,000\,\mathrm{km^2 a^{-1}}$ found here for the period from 2002 to 2017 (Figure 5b). Since the decline in SIA is accelerating (e.g. Comiso et al., 2017), the differences can be easily

explained by the different time periods used. Interestingly the range between the products with the smallest ($69\,100\,\mathrm{km^2 a^{-1}}$, NASA Team 1) and largest decline ($85\,800\,\mathrm{km^2 a^{-1}}$, HadISST2) in minimum SIA in Comiso et al. (2017) is $16\,700\,\mathrm{km^2 a^{-1}}$, or about 20% of the average estimate. This range of uncertainty is fully consistent with our single product ensemble uncertainty.

The work presented here looks at one time period, 2015 for a continuous time-series and 2002-2017 for the September trend analysis, for a specific SIC dataset (SICCI2-50km). It demonstrates how error estimates can be supplied for SIE and SIA

estimates and for their trends. In an next step our method can readily be implemented for sea-ice indicators on a daily basis by operational services such as the EUMETSAT OSI SAF. In addition to providing error estimates on their daily or monthly mean SIE and SIA time-series, it would allow the data providers to set confidence intervals for widely used metrics such as the trends in monthly SIE and SIA (typically September and March), as well as rank values such as record low/high, earliest/latest sea-ice extreme, etc., thus increasing the maturity of these key climate indicators. We further anticipate this work to inspire the

development of more sophisticated SIC error correlation estimates to refine SIA and SIE uncertainty estimates and broaden the applicability of the SIC ensemble from different retrieval algorithms. If regional error characteristics are sufficiently well represented, the SIC ensemble could be used directly in regional coupled models to investigate the impact of correlated SIC uncertainties on oceanic and atmospheric surface fluxes.

## 5 Conclusions

An analysis of errors in the CCI passive microwave SIC product indicates typical error correlations centered around nearly 300 km in space (based on the findings of Kern et al., 2021) and about 5 days in time. We derive a SIC error ensemble by statistical modelling and show that this is able to represent the local SIC uncertainty estimates, as provided by the CCI SIC product, as well as the analysed error correlations. These correlations are shown to have a strong impact on the error propagation from local SIC to aggregated SIA and SIE; The SIA uncertainty in 2015 is $306\,000\,\mathrm{km^2}$ for daily, $275\,000\,\mathrm{km^2}$

weekly, and $164\,000\,\mathrm{km^2}$ for monthly estimates. The SIE uncertainty in 2015 is slightly smaller with $296\,000\,\mathrm{km^2}$ for daily, $261\,000\,\mathrm{km^2}$ weekly, and $156\,000\,\mathrm{km^2}$ for monthly estimates. Weekly SIA and SIE averages have very similar uncertainties as daily estimates due to the temporal error correlation. These uncertainties are about half of the spread in SIA and SIE estimates from different products, which do, however, differ not only by random errors as sampled here, but also conceptional factors such as different land masks (Kern et al., 2019) or the treatment of the polar gap in satellite data. These conceptional factors

often result in biases. On the other hand, uncertainties in SIE due the algorithm parameter sensitivity as found by Meier and Stewart (2019) (of order $50\,000\,\mathrm{km^2\ a^{-1}}$) do not represent aspects like gridding and sensor noise. The uncertainties provided here originate from a single SIC product and encompass algorithmic and smearing uncertainties due to satellite footprint




mismatches, which makes them a more appropriate estimate when, e.g. comparing model and observational products on the same grid.

The Arctic September SIA trend for the period from 2002 to 2017 is estimated to be $105\,000$ km$^2$ a$^{-1}$ $\pm 9\,000$ km$^2$ a$^{-1}$. This trend is an important indicator for the sensitivity of the Arctic ocean to climate change and a good example to illustrate the strength of our approach: Biases (not represented here) are not an issue for trend analysis but our representation of temporal error correlations allows us to give a more complete picture of the trend uncertainty.

By using a simple (spatial- and temporal-) distance based correlation model to propagate the uncertainties of the underlying SIC fields to uncertainties of the key climate indicators SIA and SIE and their trends, we have been able to further our understanding and quantification of these uncertainties. We expect this quantification of observational uncertainties to be essential for our understanding of the ongoing Arctic climate change, both as a means in themselves but also in providing a more robust basis for model evaluation studies.

*Code and data availability.* The original SIC data is available from the ESA CCI website and the CEDA Archive (Pedersen et al., 2017). The code to create ensemble members based on this data is available at Wernecke (2022)

## Appendix A: Sensitivities

The SIA and SIE uncertainty estimates are strongly dependent on the error correlation length scales. While we attempt to constrain the error correlation length as well as possible, there is some ambiguity in the best representation of these correlations. To investigate the impact of uncertainties in the error correlations we test the sensitivities towards several aspect of the processing.

### A1 Sensitivity to individual correlations

First we set the spatial (temporal) error correlation to zero and use our best estimate for the temporal (spatial) error correlation. In this way we separate the impacts of the spatial and temporal correlations from each other ( Figure A1). We find that the spatial error correlation influences the magnitude of the SIA and SIE uncertainties while the temporal correlation reduces the rate at which the uncertainties reduce with temporal averaging.

### A2 Sensitivity to correlation length scales

Secondly we define error correlations on the lower and upper end of consistency with observations. Based on Figure 3 we choose $pm50$ km and $\pm 1$ day as a reasonable parameter range. We repeat the SIA and SIE uncertainty calculations for the combination of lower-end spatial and temporal error correlations as well as the combination of upper-end spatial and temporal error correlations in Figure A2. These two setups can be understood as an envelope of SIA and SIE uncertainty estimates.



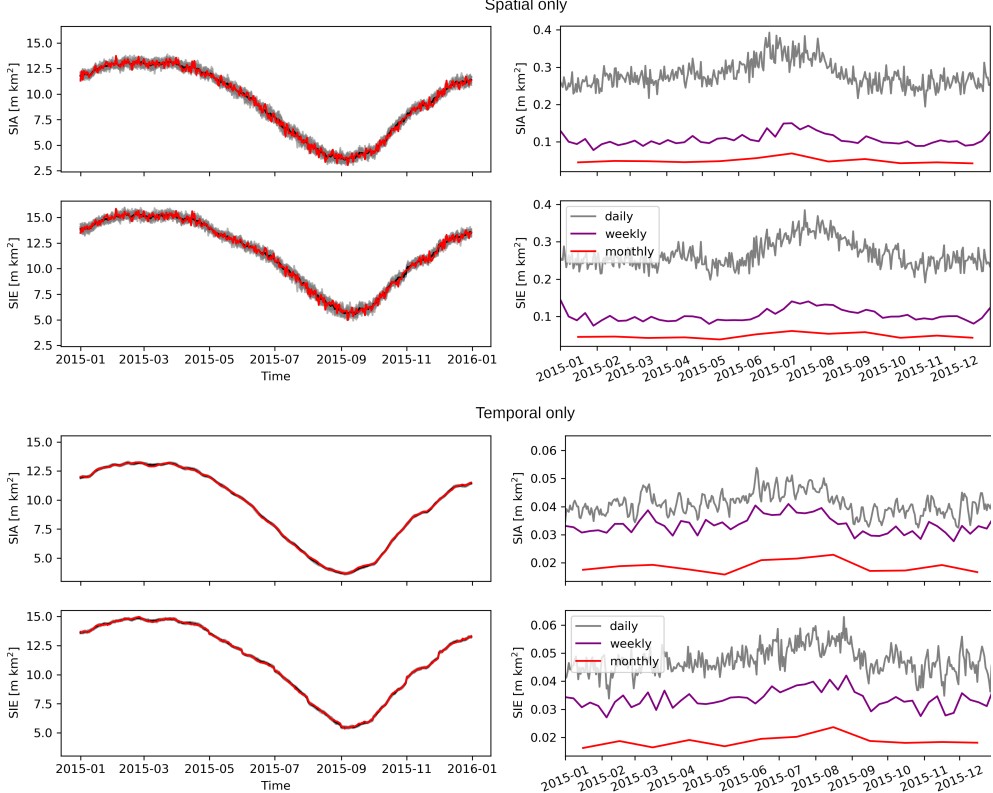

**Figure A1.** Daily Arctic SIA and SIE ensemble of 20 SIC ensemble members with one member highlighted (red) and mean (black) (left) and standard deviations of SIA and SIE derived from an ensemble of 100 SIC ensemble members (right) with spatial error correlation only (top) and temporal correlation only (bottom).

## A3   Sensitivity to filter type

To assess the sensitivity to the filter type used, we use two alternative filters to create the noise ensemble: A Fast Fourier Transform (FFT) and a wavelet filter (Xu et al., 1994). The FFT filter transforms the independent noise field into a frequency representation in which we set all frequency contributions outside a given range to zero. The inverse transformation creates the required noise in the space/time domain. The FFT is a global transformation, meaning that oscillating components in the whole noise field are preserved if they have a frequency which is not filtered out. This is important because it means that FFT bandpass filters have the tendency to create negative correlations at specific distances in addition to the desired positive correlation at small distances. Such anticorrelation are not expected to play a major role in SIC products.

A wavelet transformation is a multi-resolution decomposition of an n-dimensional image. The basis functions (wavelets) are, in contrast to the FFT, diminishing with distance to their center and are therefore supported only on local subsections of the image. Wavelet transforms are able to reveal the frequency content of the signal around a specific location which makes them attractive to identify edges in noisy images (e.g. Xu et al., 1994). Differently sized wavelets, representing different frequencies,



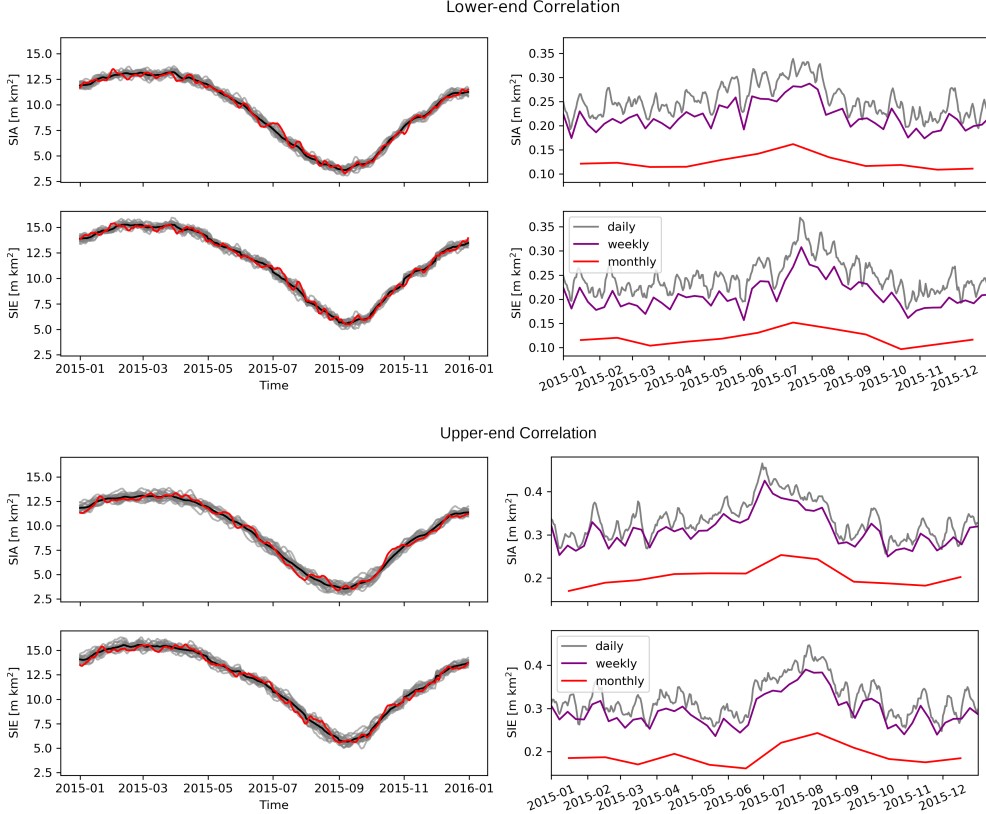

**Figure A2.** Daily Arctic SIA and SIE ensemble of 20 SIC ensemble members with one member highlighted (red) and mean (black) (left) and standard deviations of SIA and SIE derived from an ensemble of 100 SIC ensemble members (right) with lower-end (top) and and upper-end (bottom) spatial and temporal error correlations.

are used as basis function for a wavelet decomposition, providing coefficients which illustrate where, which frequency is found in the signal. Wavelet recompositions have been used before to create geophysical noise in applications similar to ours (Castleman et al., 2022).

The filtering steps are the same for FFT and wavelet filters: First the three dimensional white noise field is decomposed into its frequency components. Then frequency components/wavelet coefficients outside of a manually defined window are set to zero, followed by a reverse transformation/recomposition into the original space-time domain. For the FFT band-pass filter, the range is set to 233 km to 333 km and 4 day to 6 days. The wavelet filter decomposes the noise field into four levels using a discrete wavelet transform with Daubechies-12 wavelets (function *wavedecn()* of the python module *pywt*) and removes
the contributions from the two smallest levels before recomposition. We have set the these variables to reproduce spatial and temporal error correlation length scales as closely as possible (Figure A4) to focus on the sensitivity of the functional form of the error correlations itself. Figure A3 and Table A1 show that the sensitivity of the SIA and SIE uncertainties to the used filter type is up to 93 000 km$^2$. That being said, the FFT filter noise used here has slightly shorter spatial correlations (Figure A4).





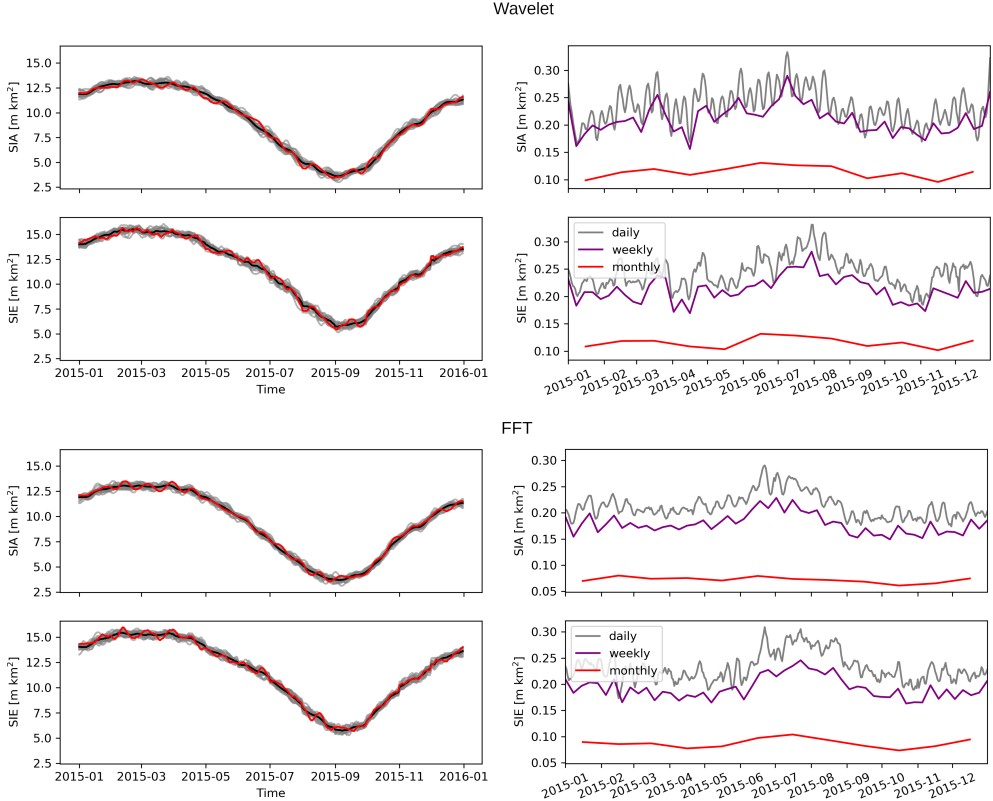

**Figure A3.** Daily Arctic SIA and SIE ensemble of 20 SIC ensemble members with one member highlighted (red) and mean (black) (left) and standard deviations of SIA and SIE derived from an ensemble of 100 SIC ensemble members (right) with based on a wavelet filter (top) and an Fast Fourier transform filter (bottom).

As we have seen in Figure A1, this leads to a smaller total SIA and SIE uncertainty, which is also what we see when comparing

Figure A3 with Figure 5a. Therefore the quoted filter type sensitivity is likely to be overestimated and, in fact, smaller than the sensitivity to the correlation length scales.

*Acknowledgements.* Thomas Lavergne is supported by the ESA Climate Change Initiative Sea Ice project (Contract 4000126449/19/I-NB)



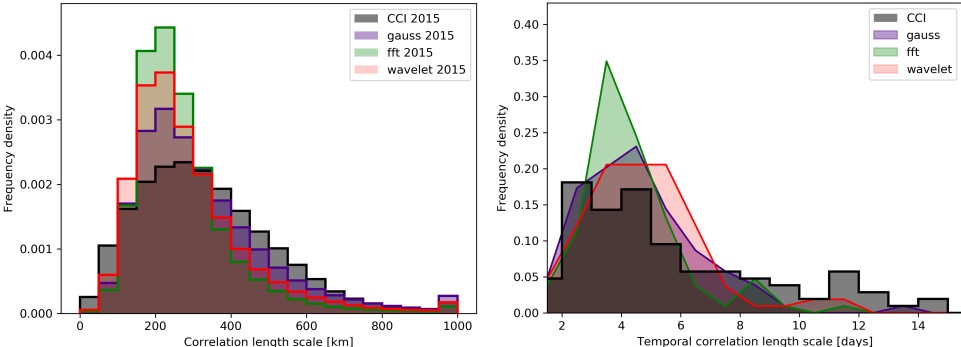

**Figure A4.** As Figure 3 but including Wavelet filter and Fast Fourier transform filter based noise characteristics.

**Table A1.** Uncertainty estimates (one STD) of 100 member ensemble for the year 2015 based on different processing types. The spatial (Sp.) and temporal (Tmp.) error correlations are nominal values and do not necessarily correspond to the averaged analysed error correlations (see text)

| Quantity | Filter | Sp-corr [km] | Tmp-corr [days] | Daily [$10^3$ km$^2$] | Weekly [$10^3$ km$^2$] | Monthly [$10^3$ km$^2$] |
|---|---|---|---|---|---|---|
| SIA | Gauss | 288 | 5 | 306 | 275 | 164 |
| SIA | Gauss | 238 | 4 | 252 | 219 | 125 |
| SIA | Gauss | 338 | 6 | 334 | 311 | 204 |
| SIA | Gauss | 288 | 0 | 280 | 106 | 49 |
| SIA | Gauss | 0 | 5 | 41 | 34 | 19 |
| SIA | FFT | 288 | 5 | 213 | 179 | 72 |
| SIA | Wavelet | 288 | 5 | 233 | 211 | 114 |
| SIE | Gauss | 288 | 5 | 296 | 261 | 156 |
| SIE | Gauss | 238 | 4 | 245 | 209 | 120 |
| SIE | Gauss | 338 | 6 | 322 | 295 | 190 |
| SIE | Gauss | 288 | 0 | 266 | 103 | 48 |
| SIE | Gauss | 0 | 5 | 47 | 34 | 19 |
| SIE | FFT | 288 | 5 | 232 | 193 | 87 |
| SIE | Wavelet | 288 | 5 | 242 | 213 | 116 |

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
