# Peer review of "Estimating the uncertainty of sea-ice area and sea-ice extent from satellite retrievals"

_EGUsphere, 2022_

## Author Comment (AC1)

Referee comments in black, response in blue

**Anonymous Referee #1:** (https://doi.org/10.5194/egusphere-2022-1189-RC1)

This study retrieved the SIA/SIE uncertainties by taking into account the spatial and temporal error correlations of the underlying local sea ice concentration products. It shows that random SIC errors play a role in SIA uncertainties comparable to inter-SIC-product biases. This study also compiles the September SIA trend with an explicit representation of temporal error correlations from 2002 to 2017. Extending this research to regional studies would help to investigate the impact of the correlated SIC uncertainties on oceanic and atmospheric surface fluxes. I would suggest a minor revision before it can be published.

Thank you for the useful comments and suggestions.

Thank you also for the suggestion of regional studies. We decided not to include those in the current manuscript. The focus of this manuscript is to showcase our approach of propagating local SIC uncertainties to SIA and SIE uncertainties as well as to analyse the main sensitivities of this approach. The results from 2015 and September are a proof of concept, and as such primarily meant to highlight the usefulness of our approach and to give a basic first idea of the resulting values. We do not aim at providing a full analysis of the entire dataset in this study. We are currently working on such analysis which we aim to publish, including regional aspects, separately once finished.

Investigating the impact of correlated SIC errors on surface fluxes, compared to independent noise, is another very interesting topic but would require modeling of convective processes in the atmosphere and ocean. It is also outside of the scope of this study.

(1) It would be better to use the exact values of SIE and SIA uncertainties in the abstract rather than showing an approximate value.

We intended to compress the information in the abstract to highlight the main findings by combining daily and weekly estimates as well as SIA and SIE estimates. However, we now provide the exact values as requested to put the reader into the position to assess the level of similarity themselves.

(2) Please indicate why you are focusing on CCI SIC product at 50km grid spacing product instead of that at 25km grid spacing.

The main difference between the 50km product and the 25km product are the used frequency channels (18.7 GHz and 37 GHz for 25km and 6.9 GHz and 37 GHz for 50km). The 6.9 GHz channel is less sensitive to some sources of uncertainty, such as atmospheric effects. Since we here are only interested in accumulated quantities (SIA and SIE), the higher spatial resolution is of limited use for this application. While we cannot say, or show here, that the 50km product is indeed more suited to derive SIA and SIE, these considerations let us focus on the 50km product for now. This product is also easier and faster to process.

(3) Please further demonstrate the advantages of using Monte Carlo approach in this study.

In general, the main advantage of MC approaches is their flexibility and user friendliness. This flexibility helps for example with the calculation of SIE uncertainty where traditional error propagation does not work well with discontinuities such as the application of thresholds.

For our purposes, however, the main reason for using an MC approach is that a traditional analytical approach is not feasible due to the size of the dataset. This is mainly because we allow temporal and spatial correlations at the same time. Therefore the number of data points handled at one time is very large, making it practically impossible to numerically define a covariance/correlation matrix (which has $n^2$ entries, where n is the number of data points). While there are approaches for sparse correlation matrices (as we have here), they are not always applicable, require additional assumptions and are only approximations.

We now discuss this impracticality of traditional analytical uncertainty propagation and the advantages of MC approaches in more detail.

(4) Why was 2015 chosen as a case to demonstrate the error correlation length scale and SIA/SIE standard deviation?

This choice was somewhat arbitrary, but was motivated by the fact that 2015 closely resembles the average SIE of the years 2011 to 2020 (https://nsidc.org/arcticseaicenews/charctic-interactive-sea-ice-graph/) and can thus be considered a "standard" year. Nevertheless, other years could have been chosen as case study since this choice only acts as an example.

(5) It would be better to add more figures to make the Results and Discussion sections more convincing and easier to understand, e.g., superimposing a graph of SIA trend and the linear regression of CCI product in Figure 5(b).

A figure has been added as suggested. In addition we added a panel with SIA and SIE anomalies to former Figure 4 and restructured the figures into one set (former Figure 4 and 5a) concerned with the 2015 time series and another set (a new figure with the September SIA plus regression and Figure 5b) concerned with the September trends. In this way we think that the result section has become more convincing and easier to follow.

---

## Author Comment (AC2)

Referee comments in black, response in blue

**Anonymous Referee #2: (https://doi.org/10.5194/egusphere-2022-1189-RC2)**

**Review of "Estimating the uncertainty of sea-ice area and sea-ice extent from satellite retrievals" by Wernecke et al.**

**Summary**

This paper presents a method to estimate uncertainties in passive microwave-derives sea ice extent and area. It is derived from spatial and temporal errors in the gridded concentration fields. The approach yields estimates of uncertainty in daily and monthly extent and area values and the paper provides trend estimates with accompanying uncertainties.

**General Comment**

This is an excellent paper. It provides a logical, quantitative method to estimate extent and area uncertainties based on the characteristics of the gridded sea ice concentration fields. Such quantitative extent and area uncertainties have long been lacking, which is a significant limitation in the passive microwave products that are a key indicator of warming and climate change. The use of these extent and area uncertainties to derive uncertainties in trends is also highly valuable, particularly for the Antarctic where trends are near-zero and uncertainty is needed to assess if trends are significant. The paper is well-written and explains the methods and results well. I recommend publication after only minor revisions, noted below.

Thank you for the review and suggestions. We are very glad to hear your thoughts on the research topic, its significance and the presentation of this study.

**Specific Comments** (by line number):

79: "tie points" is used here, but not defined. It is defined later in the paper in lines 244-245. Readers may not be familiar with the term, so it should be defined here when it is initially used.

Agreed

83: "constant biases" – aren't biases by definition a constant? I think you mean here that the biases are consistent throughout the various product – i.e., a land difference is a constant offset – as opposed to differences between products due to methodologies (channels used, tie point values) that have mean biases but with variable differences depending on conditions. I think it would be fine to just remove "constant" and just say "biases" as the source of these biases are mentioned.

Agreed

115-116: This paper essentially uses the results from Kern (2021) and Kern (2021) as the basis for the whole approach. In light of that, I think a short summary of the method and data is warranted. Though the references obviously explain things in detail, I think having a brief explanation would be helpful to allow readers to have a sense of those papers without having to go to the external references. Again, it doesn't need to be detailed, but at least 2 or 3 sentences summarizing the data and method used for both the spatial correlation (2.1.1) and temporal correlation (2.1.2) would be a good foundation for the rest of the paper. It could also be done for both spatial and temporal in Section 2.1, as an introduction, before going to the two subsections.

In parts this might be a misunderstanding, potentially caused by an ambiguous formulation on our side (line 116 'In this study' was intended to refer to Kern 2021, not the manuscript presented here). The majority of Section 2.1.1 (lines 115 to 144) is a summary and discussion of the Kern (2021) method and dataset. The spatial correlation length scale is based directly on the published data (Kern, 2022), the temporal correlation length scale is derived here, closely following the methods of Kern (2021). Both of these aspects have been made clearer and a few key results from Kern (2021) have been added.

221-229: It is most useful to have trend values with the quantitative uncertainties derived based on the spread of the ensemble members. This provides the trend uncertainties based on the uncertainty in the extent and area values. However, there is also the significance of the trend based on the "noise" in the linear trend fit – e.g., the trend standard deviation and/or the P-value of the trend (e.g., $P<0.05$); this assesses the confidence level in the trend based on the length of the timeseries and the year-to-year variability. This is the number often calculated and quoted with trends. But that is different than your estimate based on the ensemble members. I think it would be worth making this clear and perhaps it would warrant a short discussion (maybe in Section 4 or 5) of what this means for understanding the trend significance. This is particularly key for the Antarctic where trends have been near-zero, but have varied between small positive and small negative trends – are these changes really significant given what you have shown about the uncertainties as well as the trend standard deviation values?

While we have no answer to the significance of Antarctic trends for now, we do now discuss the different estimates in more detail and mention a limitation of the traditionally used standard error of the trend, namely its implicit assumption of independence.

We note that the current study focuses on Arctic sea ice and that an adaptation to the southern hemisphere is in preparation. In addition, the focus of this manuscript is to showcase our approach of propagating local SIC uncertainties to SIA and SIE uncertainties as well as to analyse the main sensitivities of this approach. We do not aim at providing a full analysis of the entire dataset in this study.